# Identifying Analogies Across Domains

**Yedid Hoshen**[1] **and Lior Wolf**[1,2]
[1]Facebook AI Research
[2]Tel Aviv University

## Abstract

Identifying analogies across domains without supervision is an important task for artificial intelligence. Recent advances in cross domain image mapping have concentrated on translating images across domains. Although the progress made is impressive, the visual fidelity many times does not suffice for identifying the matching sample from the other domain. In this paper, we tackle this very task of finding exact analogies between datasets i.e. for every image from domain A find an analogous image in domain B. We present a matching-by-synthesis approach: AN-GAN, and show that it outperforms current techniques. We further show that the cross-domain mapping task can be broken into two parts: domain alignment and learning the mapping function. The tasks can be iteratively solved, and as the alignment is improved, the unsupervised translation function reaches quality comparable to full supervision.

## 1 Introduction

Humans are remarkable in their ability to enter an unseen domain and make analogies to the previously seen domain without prior supervision ("This dinosaur looks just like my dog Fluffy"). This ability is important for using previous knowledge in order to obtain strong priors on the new situation, which makes identifying analogies between multiple domains an important problem for Artificial Intelligence. Much of the recent success of AI has been in supervised problems, i.e., when explicit correspondences between the input and output were specified on a training set. Analogy identification is different in that no explicit example analogies are given in advance, as the new domain is unseen.

Recently several approaches were proposed for unsupervised mapping between domains. The approaches take as input sets of images from two different domains $A$ and $B$ without explicit correspondences between the images in each set, e.g. Domain A: a set of aerial photos and Domain B: a set of Google-Maps images. The methods learn a mapping function $T_{AB}$ that takes an image in one domain and maps it to its likely appearance in the other domain, e.g. map an aerial photo to a Google-Maps image. This is achieved by utilizing two constraints: (i) Distributional constraint: the distributions of mapped A domain images ($T_{AB}(x)$) and images of the target domain B must be indistinguishable, and (ii) Cycle constraint: an image mapped to the other domain and back must be unchanged, i.e., $T_{BA}(T_{AB}(x)) = x$.

In this paper the task of analogy identification refers to finding pairs of examples in the two domains that are related by a fixed non-linear transformation. Although the two above constraints have been found effective for training a mapping function that is able to translate between the domains, the translated images are often not of high enough visual fidelity to be able to perform exact matching. We hypothesize that it is caused due to not having exemplar-based constraints but rather constraints on the distributions and the inversion property.

In this work we tackle the problem of analogy identification. We find that although current methods are not designed for this task, it is possible to add exemplar-based constraints in order to recover high performance in visual analogy identification. We show that our method is effective also when only some of the sample images in $A$ and $B$ have exact analogies whereas the rest do not have exact analogies in the sample sets. We also show that it is able to find correspondences between sets when no exact correspondences are available at all. In the latter case, since the method retrieves rather than maps examples, it naturally yields far better visual quality than the mapping function.

Using the domain alignment described above, it is now possible to perform a two step approach for training a domain mapping function, which is more accurate than the results provided by previous unsupervised mapping approaches:

1. Find the analogies between the $A$ and $B$ domain, using our method.
2. Once the domains are aligned, fit a translation function $T_{AB}$ between the domains $y_{m_i} = T_{AB}(x_i)$ using a fully supervised method. For the supervised network, larger architectures and non-adversarial loss functions can be used.

## 2 Related work

This paper aims to identify analogies between datasets without supervision. Analogy identification as formulated in this paper is highly related to image matching methods. As we perform matching by synthesis across domains, our method is related to unsupervised style-transfer and image-to-image mapping. In this section we give a brief overview of the most closely related works.

**Image Matching** Image matching is a long-standing computer vision task. Many approaches have been proposed for image matching, most notably pixel- and feature-point based matching (e.g. SIFT (Lowe, 2004)). Recently supervised deep neural networks have been used for matching between datasets (Wang et al., 2014), and generic visual features for matching when no supervision is available (e.g. (Ganin & Lempitsky, 2015)). As our scenario is unsupervised, generic visual feature matching is of particular relevance. We show in our experiments however that as the domains are very different, standard visual features (multi-layer VGG-16 (Simonyan & Zisserman, 2015)) are not able to achieve good analogies between the domains.

**Generative Adversarial Networks** GAN (Goodfellow et al., 2014) technology presents a major breakthrough in image synthesis (and other domains). The success of previous attempts to generate random images in a class of a given set of images, was limited to very specific domains such as texture synthesis. Therefore, it is not surprising that most of the image to image translation work reported below employ GANs in order to produce realistically looking images. GAN (Goodfellow et al., 2014) methods train a generator network $G$ that synthesizes samples from a target distribution, given noise vectors, by jointly training a second network $D$. The specific generative architecture we and others employ is based on the architecture of (Radford et al., 2015). In image mapping, the created image is based on an input image and not on random noise (Kim et al., 2017; Zhu et al., 2017; Yi et al., 2017; Liu & Tuzel, 2016; Taigman et al., 2017; Isola et al., 2017).

**Unsupervised Mapping** Unsupervised mapping does not employ supervision apart from sets of sample images from the two domains. This was done very recently (Taigman et al., 2017; Kim et al., 2017; Zhu et al., 2017; Yi et al., 2017) for image to image translation and slightly earlier for translating between natural languages (Xia et al., 2016). The above mapping methods however are focused on generating a mapped version of the sample in the other domain rather than retrieving the best matching sample in the new domain.

**Supervised Mapping** When provided with matching pairs of (input image, output image) the mapping can be trained directly. An example of such method that also uses GANs is (Isola et al., 2017), where the discriminator $D$ receives a pair of images where one image is the source image and the other is either the matching target image ("real" pair) or a generated image ("fake" pair); The link between the source and the target image is further strengthened by employing the U-net architecture of (Ronneberger et al., 2015). We do not use supervision in this work, however by the successful completion of our algorithm, correspondences are generated between the domains, and supervised mapping methods can be used on the inferred matches. Recently, (Chen & Koltun, 2017) demonstrated improved mapping results, in the supervised settings, when employing the perceptual loss and without the use of GANs.

## 3 Method

In this section we detail our method for analogy identification. We are given two sets of images in domains $A$ and $B$ respectively. The set of images in domain $A$ are denoted $x_i$ where $i \in I$ and the set image in domain $B$ are denoted $y_j$ where $j \in J$. Let $m_i$ denote the index of the $B$ domain image

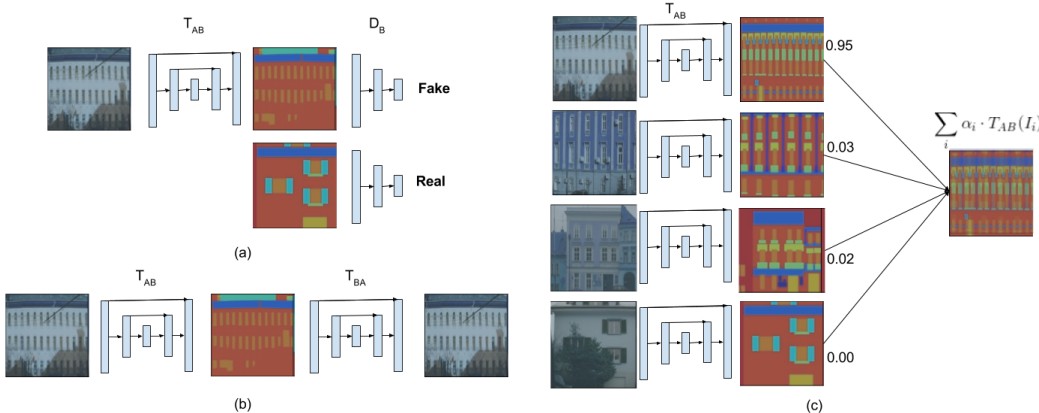

Figure 1: A schematic of our algorithm with illustrative images. (a) Distribution matching using an adversarial loss. Note that in fact this happens for both the $A \rightarrow B$ and $B \rightarrow A$ mapping directions. (b) Cycle loss ensuring that an example mapped to the other domain and back is mapped to itself. This loss is used for both $A \rightarrow B \rightarrow A$ and $B \rightarrow A \rightarrow B$. (c) Exemplar loss: The input domain images are mapped to the output domain and then a linear combination of the mapped images is computed. The objective is to recover a specific target domain image. Both the linear parameters $\alpha$ and mapping $T$ are jointly learned. The loss is computed for both $A \rightarrow B$ and $B \rightarrow A$ directions. The parameters $\alpha$ are shared in the two directions.

$y_{m_i}$ that is analogous to $x_i$. Our goal is to find the matching indexes $m_i$ for $i \in I$ in order to be able to match every $A$ domain image $x_i$ with a $B$ domain image $y_{m_i}$, if such a match exists.

We present an iterative approach for finding matches between two domains. Our approach maps images from the source domain to the target domain, and searches for matches in the target domain.

## 3.1 Distribution-based Mapping

A GAN-based distribution approach has recently emerged for mapping images across domains. Let $x$ be an image in domain $A$ and $y$ be an image in domain $B$. A mapping function $T_{AB}$ is trained to map $x$ to $T_{AB}(x)$ so that it appears as if it came from domain $B$. More generally, the distribution of $T_{AB}(x)$ is optimized to appear identical to that of $y$. The distributional alignment is enforced by training a discriminator $D$ to discriminate between samples from $p(T_{AB}(x))$ and samples from $p(y)$, where we use $p(x)$ to denote the distribution of $x$ and $p(T_{AB}(x))$ to denote the distribution of $T_{AB}(x)$ when $x \sim p(x)$. At the same time $T_{AB}$ is optimized so that the discriminator will have a difficult task of discriminating between the distributions.

The loss function for training $T$ and $D$ are therefore:

$$min_{T_{AB}} L_T = \mathcal{L}_b(D(T_{AB}(x)), 1) \tag{1}$$

$$min_D L_D = \mathcal{L}_b(D(T_{AB}(x)), 0) + \mathcal{L}_b(D(y), 1) \tag{2}$$

Where $\mathcal{L}_b(,)$ is a binary cross-entropy loss. The networks $L_D$ and $L_T$ are trained iteratively (as they act in opposite directions).

In many datasets, the distribution-constraint alone was found to be insufficient. Additional constraints have been effectively added such as circularity (cycle) (Zhu et al., 2017; Kim et al., 2017) and distance invariance (Benaim & Wolf, 2017). The popular cycle approach trains one-sided GANs in both the $A \rightarrow B$ and $B \rightarrow A$ directions, and then ensures that an $A$ image domain translated to $B$ ($T_{AB}(x)$) and back to $A$ ($T_{BA}(T_{BA}(x))$) recovers the original $x$.

Let $\mathcal{L}_1$ denote the $L_1$ loss. The complete two-sided cycle loss function is given by:

$$L_{T_{dist}} = \mathcal{L}_b(D_A(T_{BA}(y)), 1) + \mathcal{L}_b(D_B(T_{AB}(x)), 1) \tag{3}$$

$$L_{T_{cycle}} = \mathcal{L}_1(T_{AB}(T_{BA}(y)), y) + \mathcal{L}_1(T_{BA}(T_{AB}(x)), x) \tag{4}$$

$$min_{T_{AB}, T_{BA}} L_T = L_{T_{dist}} + L_{T_{cycle}} \tag{5}$$

$$min_{D_A, D_B} L_D = \mathcal{L}_b(D_A(T_{BA}(y)), 0) + \mathcal{L}_b(D_A(x), 1) + \mathcal{L}_b(D_B(T_{AB}(x)), 0) + \mathcal{L}_b(D_B(y), 1) \quad (6)$$

The above two-sided approach yields mapping function from $A$ to $B$ and back. This method provides matching between every sample and a synthetic image in the target domain (which generally does not correspond to an actual target domain sample), it therefore does not provide exact correspondences between the $A$ and $B$ domain images.

## 3.2 EXEMPLAR-BASED MATCHING

In the previous section, we described a distributional approach for mapping $A$ domain image $x$ to an image $T_{AB}(x)$ that appears to come from the $B$ domain. In this section we provide a method for providing exact matches between domains.

Let us assume that for every $A$ domain image $x_i$ there exists an analogous $B$ domain image $y_{m_i}$. Our task is find the set of indices $m_i$. Once the exact matching is recovered, we can also train a fully supervised mapping function $T_{AB}$, and thus obtain a mapping function of the quality provided by supervised method.

Let $\alpha_{i,j}$ be the proposed match matrix between $B$ domain image $y_j$ and $A$ domain image $x_i$, i.e., every $x_i$ matches a mixture of all samples in $B$, using weights $\alpha_{i,:}$, and similarly for $y_j$ for a weighing using $\alpha_{:,j}$ of the training samples from $A$. Ideally, we should like a binary matrix with $\alpha_{i,j} = 1$ for the proposed match and 0 for the rest. This task is formally written as:

$$\mathcal{L}_p(\sum_i \alpha_{i,j} T_{AB}(x_i), y_j), \quad (7)$$

where $\mathcal{L}_p$ is a "perceptual loss", which is based on some norm, a predefined image representation, a Laplacian pyramid, or otherwise. See Sec. 3.4.

The optimization is continuous over $T_{AB}$ and binary programming over $\alpha_{i,j}$. Since this is computationally hard, we replace the binary constraint on $\alpha$ by the following relaxed version:

$$\sum_i \alpha_{i,j} = 1 \quad (8)$$

$$\alpha_{i,j} \geq 0 \quad (9)$$

In order to enforce sparsity, we add an entropy constraint encouraging sparse solutions.

$$L_e = \sum_{i,j} -\alpha_{i,j} \cdot \log(\alpha_{i,j}) \quad (10)$$

The final optimization objective becomes:

$$min_{T_{AB}, \alpha_{i,j}} L_{T_{exemplar}} = \sum_j \mathcal{L}_p(\sum_i \alpha_{i,j} T_{AB}(x_i), y_j) + k_{entropy} \cdot L_e \quad (11)$$

The positivity $\alpha \geq 0$ and $\sum_i \alpha_{i,j} = 1$ constraints are enforced by using an auxiliary variable $\beta$ and passing it through a $Softmax$ function.

$$\alpha_{i,j} = Softmax_i(\beta_{i,j}) \quad (12)$$

The relaxed formulation can be optimized using SGD. By increasing the significance of the entropy term (increasing $k_{entropy}$), the solutions can converge to the original correspondence problem and exact correspondences are recovered at the limit.

Since $\alpha$ is multiplied with all mapped examples $T_{AB}(x)$, it might appear that mapping must be performed on all $x$ samples at every batch update. We have however found that iteratively updating $T_{AB}$ for $N$ epochs, and then updating $\beta$ for $N$ epochs ($N = 10$) achieves excellent results. Denote the $\beta$ (and $\alpha$) updates- $\alpha$ iterations and the updates of $T_{AB}$ - $T$ iterations. The above training scheme requires the full mapping to be performed only once at the beginning of the $\alpha$ iteration (so once in $2N$ epochs).

### 3.3 AN-GAN

Although the examplar-based method in Sec. 3.2 is in principle able to achieve good matching, the optimization problem is quite hard. We have found that a good initialization of $T_{AB}$ is essential for obtaining good performance. We therefore present AN-GAN - a cross domain matching method that uses both exemplar and distribution based constraints.

The AN-GAN loss function consists of three separate constraint types:

1. Distributional loss $L_{T_{dist}}$: The distributions of $T_{AB}(x)$ matches $y$ and $T_{BA}(y)$ matches $x$ (Eq. 3).
2. Cycle loss $L_{T_{cycle}}$: An image when mapped to the other domain and back should be unchanged (Eq. 4).
3. Exemplar loss $L_{T_{exemplar}}$: Each image should have a corresponding image in the other domain to which it is mapped (Eq. 11).

The AN-GAN optimization problem is given by:

$$\min_{T_{AB}, T_{BA}, \alpha_{i,j}} L_T = L_{T_{dist}} + \gamma \cdot L_{T_{cycle}} + \delta \cdot L_{T_{exemplar}} \tag{13}$$

The optimization also adversarially trains the discriminators $D_A$ and $D_B$ as in equation Eq. 6.

*Implementation:* Initially $\beta$ are all set to 0 giving all matches equal likelihood. We use an initial burn-in period of 200 epochs, during which $\delta = 0$ to ensure that $T_{AB}$ and $T_{BA}$ align the distribution before aligning individual images. We then optimize the examplar-loss for one $\alpha$-iteration of 22 epochs, one $T$-iteration of 10 epochs and another $\alpha$-iteration of 10 epochs (joint training of all losses did not yield improvements). The initial learning rate for the exemplar loss is $1e - 3$ and it is decayed after 20 epochs by a factor of 2. We use the same architecture and hyper-parameters as CycleGAN (Zhu et al., 2017) unless noted otherwise. In all experiments the $\beta$ parameters are shared between the two mapping directions, to let the two directions inform each other as to likelihood of matches. All hyper-parameters were fixed across all experiments.

### 3.4 LOSS FUNCTION FOR IMAGES

In the previous sections we assumed a "good" loss function for determining similarity between actual and synthesized examples. In our experiments we found that Euclidean or $L_1$ loss functions were typically not perceptual enough to provide good supervision. Using the Laplacian pyramid loss as in GLO (Bojanowski et al., 2017) does provide some improvement. The best performance was however achieved by using a perceptual loss function. This was also found in several prior works (Dosovitskiy & Brox, 2016), (Johnson et al., 2016), (Chen & Koltun, 2017).

For a pair of images $I_1$ and $I_2$, our loss function first extracts VGG features for each image, with the number of feature maps used depending on the image resolution. We use the features extracted by the the second convolutional layer in each block, 4 layers in total for 64X64 resolution images and five layers for 256X256 resolution images. We additionally also use the $L_1$ loss on the pixels to ensure that the colors are taken into account. Let us define the feature maps for images $I_1$ and $I_2$ as $\phi_1^m$ and $\phi_2^m$ ($m$ is an index running over the feature maps). Our perceptual loss function is:

$$\mathcal{L}_p(I_1, I_2) = \frac{1}{N_P} L_1(I_1, I_2) + \sum_m \frac{1}{N_m} L_1(\phi_1^m, \phi_1^m) \tag{14}$$

Where $N_P$ is the number of pixels and $N_m$ is the number of features in layer $m$. We argue that using this loss, our method is still considered to be unsupervised matching, since the features are available off-the-shelf and are not tailored to our specific domains. Similar features have been extracted using completely unsupervised methods (see e.g. (Donahue et al., 2016))

## 4 EXPERIMENTS

To evaluate our approach we conducted matching experiments on multiple public datasets. We have evaluated several scenarios: (i) Exact matches: Datasets on which all $A$ and $B$ domain images have

Table 1: $A \rightarrow B$ / $B \rightarrow A$ top-1 accuracy for exact matching.

| Method | $Facades$ | $Maps$ | $E2S$ | $E2H$ |
|---|---|---|---|---|
| $Unmapped - Pixel$ | 0.00/0.00 | 0.00/0.00 | 0.06/0.00 | 0.11/0.00 |
| $Unmapped - VGG$ | 0.13/0.00 | 0.00/0.00 | 0.09/0.06 | 0.02/0.04 |
| $CycleGAN - Pixel$ | 0.00/0.41 | 0.01/0.41 | 0.02/0.20 | 0.01/0.00 |
| $CycleGAN - VGG$ | 0.51/0.34 | 0.02/0.47 | 0.21/0.31 | 0.19/0.00 |
| $\alpha$ iterations only | 0.76/0.91 | 0.83/0.73 | 0.52/0.86 | 0.74/0.76 |
| $AN - GAN$ | **0.97/0.98** | **0.87/0.87** | **0.89/0.98** | **0.94/0.91** |

exact corresponding matches. In this task the goal is to recover all the exact matches. (ii) Partial matches: Datasets on which some $A$ and $B$ domain images have exact corresponding matches. In this task the goal is to recover the actual matches and not be confused by the non-matching examples. (iii) Inexact matches: Datasets on which $A$ and $B$ domain images do not have exact matches. In this task the objective is to identify qualitatively similar matches. (iv) Inexact point cloud matching: finding the 3D transformation between points sampled for a reference and target object. In this scenario the transformation is of dimensionality $3X3$ and the points do not have exact correspondences.

We compare our method against a set of other methods exploring the state of existing solutions to cross-domain matching:

$Unmapped - Pixel$: Finding the nearest neighbor of the source image in the target domain using $L_1$ loss on the raw pixels.

$Unmapped - VGG$: Finding the nearest neighbor of the source image in the target domain using VGG feature loss (as described in Sec. 3.4. Note that this method is computationally quite heavy due to the size of each feature. We therefore randomly subsampled every feature map to 32000 values, we believe this is a good estimate of the performance of the method.

$CycleGAN - Pixel$: Train Eqs. 5, 6 using the authors' $CycleGAN$ code. Then use $L_1$ to compute the nearest neighbor in the target set.

$CycleGAN - VGG$: Train Eqs. 5, 6 using the authors' $CycleGAN$ code. Then use VGG loss to compute the nearest neighbor in the target set. The VGG features were subsampled as before due to the heavy computational cost.

$\alpha$ iterations only: Train $AN - GAN$ as described in Sec. 3.3 but with $\alpha$ iterations only, without iterating over $T_{XY}$.

$AN - GAN$: Train $AN - GAN$ as described in Sec. 3.3 with both $\alpha$ and $T_{XY}$ iterations.

## 4.1 Exact Matching Experiments

We evaluate our method on 4 public exact match datasets:

Facades: 400 images of building facades aligned with segmentation maps of the buildings (Radim Tyleček, 2013).

Maps: The Maps dataset was scraped from Google Maps by (Isola et al., 2017). It consists of aligned Maps and corresponding satellite images. We use the 1096 images in the training set.

$E2S$: The original dataset contains around 50K images of shoes from the Zappos50K dataset (Yu & Grauman, 2014), (Yu & Grauman). The edge images were automatically detected by (Isola et al., 2017) using HED ((Xie & Tu, 2015)).

$E2H$: The original dataset contains around 137k images of Amazon handbags ((Zhu et al., 2016)). The edge images were automatically detected using HED by (Isola et al., 2017).

For both $E2S$ and $E2H$ the datasets were randomly down-sampled to 2k images each to accommodate the memory complexity of our method. This shows that our method works also for moderately sized dataset.

### 4.1.1 UNSUPERVISED EXACT MATCH EXPERIMENTS

In this set of experiments, we compared our method with the five methods described above on the task of exact correspondence identification. For each evaluation, both $A$ and $B$ images are shuffled prior to training. The objective is recovering the full match function $m_i$ so that $x_i$ is matched to $y_{m_i}$. The performance metric is the percentage of images for which we have found the exact match in the other domain. This is calculated separately for $A \rightarrow B$ and $B \rightarrow A$.

The results are presented in Table. 1. Several observations are apparent from the results: matching between the domains using pixels or deep features cannot solve this task. The domains used in our experiments are different enough such that generic features are not easily able to match between them. Simple mapping using CycleGAN and matching using pixel-losses does improve matching performance in most cases. CycleGAN performance with simple matching however leaves much space for improvement.

The next baseline method matched perceptual features between the mapped source images and the target images. Perceptual features have generally been found to improve performance for image retrieval tasks. In this case we use VGG features as perceptual features as described in Sec. 3.4. We found exhaustive search too computationally expensive (either in memory or runtime) for our datasets, and this required subsampling the features. Perceptual features performed better than pixel matching.

We also run the $\alpha$ iterations step on mapped source domain images and target images. This method matched linear combinations of mapped images rather than a single image (the largest $\alpha$ component was selected as the match). This method is less sensitive to outliers and uses the same $\beta$ parameters for both sides of the match ($A \rightarrow B$ and $B \rightarrow A$) to improve identification. The performance of this method presented significant improvements.

The exemplar loss alone should in principle recover a plausible solution for the matches between the domains and the mapping function. However, the optimization problem is in practice hard and did not converge. We therefore use a distributional auxiliary loss to aid optimization. When optimized with the auxiliary losses, the exemplar loss was able to converge through $\alpha - T$ iterations. This shows that the distribution and cycle auxiliary losses are essential for successful analogy finding.

Our full-method AN-GAN uses the full exemplar-based loss and can therefore optimize the mapping function so that each source sample matches the nearest target sample. It therefore obtained significantly better performance for all datasets and for both matching directions.

### 4.1.2 PARTIAL EXACT MATCHING EXPERIMENTS

In this set of experiments we used the same datasets as above but with $M\%$ of the matches being unavailable This was done by randomly removing images from the A and B domain datasets. In this scenario $M\%$ of the domain $A$ samples do not have a match in the sample set in the $B$ domain and similarly $M\%$ of the $B$ images do not have a match in the $A$ domain. $(1-M)\%$ of $A$ and $B$ images contain exact matches in the opposite domain. The task is identification of the correct matches for all the samples that possess matches in the other domain. The evaluation metric is the percentage of images for which we found exact matches out of the total numbers of images that have an exact match. Apart from the removal of the samples resulting in $M\%$ of non-matching pairs, the protocol is identical to Sec. 4.1.1.

The results for partial exact matching are shown in Table. 2. It can be clearly observed that our method is able to deal with scenarios in which not all examples have matches. When 10% of samples do not have matches, results are comparable to the clean case. The results are not significantly lower for most datasets containing 25% of samples without exact matches. Although in the general case a low exact match ratio lowers the quality of mapping function and decreases the quality of matching, we have observed that for several datasets (notably Facades), AN-GAN has achieved around 90% match rate with as much as 75% of samples not having matches.

### 4.2 INEXACT MATCHING EXPERIMENT

Although the main objective of this paper is identifying exact analogies, it is interesting to test our approach on scenarios in which no exact analogies are available. In this experiment, we qualitatively

Table 2: Matching (top-1) accuracy for both directions $A \rightarrow B$ / $B \rightarrow A$ where $M\%$ of the examples do not have matches in the other domain. This is shown for a method that performs only $\alpha$ iterations and for the full $AN - GAN$ method.

| Experiment | | Dataset | | | |
|---|---|---|---|---|---|
| $M\%$ | Method | $Facades$ | $Maps$ | $E2S$ | $E2H$ |
| 0% | $\alpha$ iteration only | 0.76/0.91 | 0.83/0.73 | 0.52/0.86 | 0.74/0.76 |
| 0% | $AN - GAN$ | 0.97/0.98 | 0.87/0.87 | 0.89/0.98 | 0.94/0.91 |
| 10% | $\alpha$ iteration only | 0.86/0.71 | 0.79/0.66 | 0.73/0.81 | 0.85/0.75 |
| 10% | $AN - GAN$ | 0.96/0.96 | 0.86/0.88 | 0.99/0.99 | 0.93/0.84 |
| 25% | $\alpha$ iteration only | 0.73/0.70 | 0.77/0.67 | 0.66/0.87 | 0.83/0.87 |
| 25% | $AN - GAN$ | 0.93/0.94 | 0.82/0.84 | 0.84/0.81 | 0.83/0.87 |

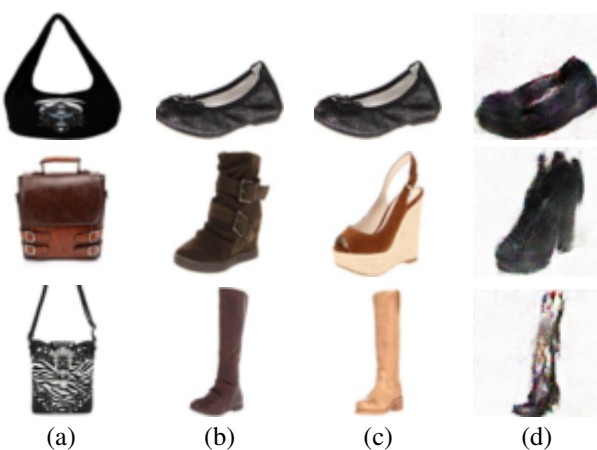

| (a) | (b) | (c) | (d) |

Figure 2: Inexact matching scenarios: Three examples of Shoes2Handbags matching. (a) Source image. (b) AN-GAN analogy. (c) DiscoGAN + $\alpha$ iterations. (d) output of mapping by DiscoGAN.

evaluate our method on finding similar matches in the case where an exact match is not available. We evaluate on the Shoes2Handbags scenario from (Kim et al., 2017). As the CycleGAN architecture is not effective at non-localized mapping we used the DiscoGAN architecture (Kim et al., 2017) for the mapping function (and all of the relevant hyper-parameters from that paper).

In Fig. 2 we can observe several analogies made for the Shoes2Handbags dataset. The top example shows that when DiscoGAN is able to map correctly, matching works well for all methods. However in the bottom two rows, we can see examples that the quality of the DiscoGAN mapping is lower. In this case both the DiscoGAN map and DiscoGAN + $\alpha$ iterations present poor matches. On the other hand $AN - GAN$ forced the match to be more relevant and therefore the analogies found by $AN - GAN$ are better.

### 4.3 Applying a supervised method on top of AN-GAN's matches

We have shown that our method is able to align datasets with high accuracy. We therefore suggested a two-step approach for training a mapping function between two datasets that contain exact matches but are unaligned: (i) Find analogies using $AN - GAN$, and (ii) Train a standard mapping function using the self-supervision from stage (i).

For the Facades dataset, we were able to obtain 97% alignment accuracy. We used the alignment to train a fully self-supervised mapping function using Pix2Pix (Isola et al., 2017). We evaluate on the facade photos to segmentations task as it allows for quantitative evaluation. In Fig. 3 we show two facade photos from the test set mapped by: CycleGAN, Pix2Pix trained on AN-GAN matches

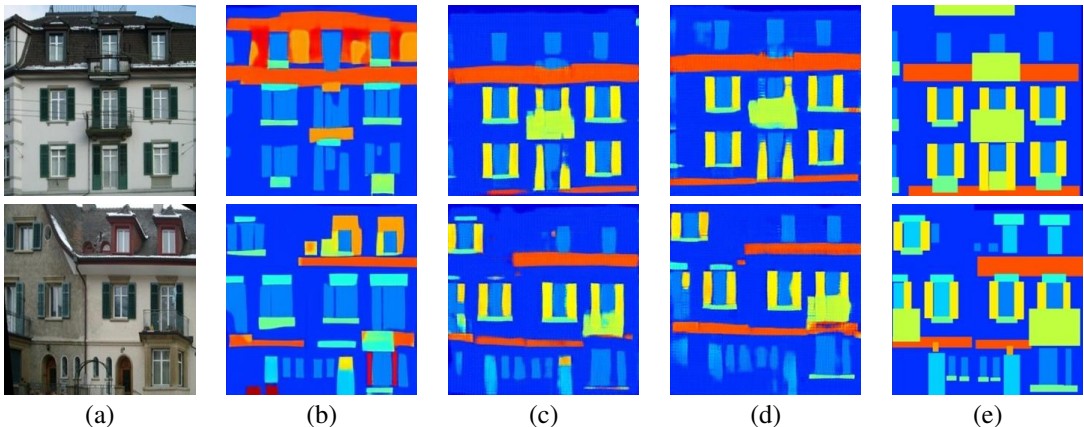

|      (a)      |      (b)      |      (c)      |      (d)      |      (e)      |

Figure 3: Supervised vs unsupervised image mapping: The supervised mapping is far more accurate than unsupervised mapping, which is often unable to match the correct colors (segmentation labels). Our method is able to find correspondences between the domains and therefore makes the unsupervised problem, effectively supervised. (a) Source image. (b) Unsupervised translation using CycleGAN. (c) A one to one method (Pix2Pix) as trained on AN-GAN's matching, which were obtained in an unsupervised way. (d) Fully supervised Pix2Pix mapping using correspondences. (e) Target image.

Table 3: Facades $\rightarrow$ Labels. Segmentation Test-Set Accuracy.

| Method | Per-Pixel Acc. | Per-Class Acc. | Class IOU |
|---|---|---|---|
| $CycleGAN$ | 0.425 | 0.252 | 0.163 |
| $Pix2Pix$ with $An-GAN$ matches | **0.535** | **0.366** | **0.258** |
| $Pix2Pix$ with supervision | **0.531** | **0.358** | **0.250** |

and a fully-supervised Pix2Pix approach. We can see that the images mapped by our method are of higher quality than CycleGAN and are about the fully-supervised quality. In Table. 3 we present a quantitative comparison on the task. As can be seen, our self-supervised method performs similarly to the fully supervised method, and much better than CycleGAN.

We also show results for the *edges2shoes* and *edges2handbags* datasets. The supervised stage uses a Pix2Pix architecture, but only $L_1$ loss (rather than the combination with cGAN as in the paper – $L_1$ only works better for this task). The test set $L_1$ error is shown in Tab. 4. It is evident that the use of an appropriate loss and larger architecture enabled by the ANGAN-supervision yields improved performance over CycleGAN and is competitive with full-supervision.

### 4.4 POINT CLOUD MATCHING

We have also evaluated our method on point cloud matching in order to test our method in low dimensional settings as well as when there are close but not exact correspondences between the samples in the two domains. Point cloud matching consists of finding the rigid 3D transformation between a set of points sampled from the reference and target 3D objects. The target 3D object is a

Table 4: Edge Prediction Test-Set $L_1$ Error.

| Dataset | Edges2Shoes | Edges2Handbags |
|---|---|---|
| $CycleGAN$ | 0.049 | 0.077 |
| $Pix2Pix$ with $An-GAN$ matches | **0.039** | **0.056** |
| $Pix2Pix$ with supervision | **0.039** | **0.057** |

Table 5: Evaluation of the point cloud alignment success probability of our method and CycleGAN

| $Rotation angle$ | CycleGAN | $Ours$ |
|---|---|---|
| 0-30 | 0.12000 | 1.00000 |
| 30-60 | 0.12500 | 1.00000 |
| 60-90 | 0.11538 | 0.88462 |
| 90-120 | 0.07895 | 0.78947 |
| 120-150 | 0.05882 | 0.64706 |
| 150-180 | 0.10000 | 0.76667 |

transformed version of the model and the sampled points are typically not identical between the two point clouds.

We ran the experiments using the Bunny benchmark, using the same setting as in (Vongkulbhisal et al., 2017). In this benchmark, the object is rotated by a random 3D rotation, and we tested the success rate of our model in achieving alignment for various ranges of rotation angles. For both CycleGAN and our method, the following architecture was used. $D$ is a fully connected network with 2 hidden layers, each of $2048$ hidden units, followed by BatchNorm and with Leaky ReLU activations. The mapping function is a linear affine matrix of size $3X3$ with a bias term. Since in this problem, the transformation is restricted to be a rotation matrix, in both methods we added a loss term that encourages orthonormality of the weights of the mapper. Namely, $\|WW^T - I\|$, where $W$ are the weights of our mapping function.

Tab. 5 depicts the success rate for the two methods, for each rotation angle bin, where success is defined in this benchmark as achieving an RMSE alignment accuracy of $0.05$.

Our results significantly outperform the baseline results reported in (Vongkulbhisal et al., 2017) at large angles. Their results are given in graph form, therefore the exact numbers could not be presented in Tab. 5. Inspection of the middle column of Fig.3 in (Vongkulbhisal et al., 2017) will verify that our method performs the best for large transformations. We therefore conclude that our method is effective also for low dimensional transformations and well as settings in which exact matches do not exist.

## 5 CONCLUSION

We presented an algorithm for performing cross domain matching in an unsupervised way. Previous work focused on mapping between images across domains, often resulting in mapped images that were too inaccurate to find their exact matches. In this work we introduced the exemplar constraint, specifically designed to improve match performance. Our method was evaluated on several public datasets for full and partial exact matching and has significantly outperformed baseline methods. It has been shown to work well even in cases where exact matches are not available. This paper presents an alternative view of domain translation. Instead of performing the full operation end-to-end it is possible to (i) align the domains, and (ii) train a fully supervised mapping function between the aligned domains.

Future work is needed to explore matching between different modalities such as images, speech and text. As current distribution matching algorithms are insufficient for this challenging scenario, new ones would need to be developed in order to achieve this goal.

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
