# OpenReview forum: "Identifying Analogies Across Domains"
_ICLR.cc/2018/Conference — Accept (Poster)_

### Official Review · AnonReviewer3 · 2017-11-29
**AN-GAN: match-aware translation of images across domains, new ideas for combining image matching and GANs**

**Rating:** 7
**Confidence:** 4

**Review:**

This paper presents an image-to-image cross domain translation framework based on generative adversarial networks. The contribution is the addition of an explicit exemplar constraint into the formulation which allows best matches from the other domain to be retrieved. The results show that the proposed method is superior for the task of exact correspondence identification and that AN-GAN rivals the performance of pix2pix with strong supervision.


Negatives:
1.) The task of exact correspondence identification seems contrived. It is not clear which real-world problems have this property of having both all inputs and all outputs in the dataset, with just the correspondence information between inputs and outputs missing.
2.) The supervised vs unsupervised experiment on Facades->Labels (Table 3) is only one scenario where applying a supervised method on top of AN-GAN’s matches is better than an unsupervised method.  More transfer experiments of this kind would greatly benefit the paper and support the conclusion that “our self-supervised method performs similarly to the fully supervised method.”

Positives:
1.) The paper does a good job motivating the need for an explicit image matching term inside a GAN framework
2.) The paper shows promising results on applying a supervised method on top of AN-GAN’s matches.

Minor comments:
1. The paper sometimes uses L1 and sometimes L_1, it should be L_1 in all cases.
2. DiscoGAN should have the Kim et al citation, right after the first time it is used. I had to look up DiscoGAN to realize it is just Kim et al.

---

> ### Author Response · Authors · 2017-12-13
> **Response**
>
> We thank you for highlighting the novelty and successful motivation of the exemplar-based matching loss.
>
> We think that the exact-analogy problem is very important.  Please refer to our comment to AnonReviewer2 for an extensive discussion.
>
> Following your request, we have added AN-GAN supervised experiments for the edges2shoes and edges2handbags datasets. The results as for the Facades case are very good.
>
> Thank you for highlighting the inconsistency in L_1 notation and the confusing reference. This has been fixed in the revised version.

---

### Official Review · AnonReviewer1 · 2017-11-30
**The approach is interesting but the paper lacks clarity of presentation**

**Rating:** 5
**Confidence:** 4

**Review:**

The paper presents a method for finding related images (analogies) from different domains based on matching-by-synthesis. The general idea is interesting and the results show improvements over previous approaches, such as CycleGAN (with different initializations, pre-learned or not). The algorithm is tested on three datasets.

While the approach has some strong positive points, such as good experiments and theoretical insights (the idea to match by synthesis and the proposed loss which is novel, and combines the proposed concepts), the paper lacks clarity and sufficient details.

Instead of the longer intro and related work discussion, I would prefer to see a Figure with the architecture and more illustrative examples to show that the insights are reflected in the experiments. Also, the matching part, which is discussed at the theoretical level, could be better explained and presented at a more visual level. It is hard to understand sufficiently well what the formalism means without more insight.

Also, the experiments need more details. For example, it is not clear what the numbers in Table 2 mean.

---

> ### Author Response · Authors · 2017-12-13
> **Response**
>
> Thank you for your positive feedback on the theoretical and experimental merits of this paper.
>
> Following your feedback on the clarity of presentation of the method. we included a diagram (including example images) illustrating the algorithm. To help keep the length under control, we shortened the introduction and related work section as you suggested.
>
> We further clarified the text of the experiments. Specifically the numbers in Tab 2 are the top-1 accuracy for both directions (A to B and B to A) when 0%, 10% and 25% of examples do not have matches in the other domain. If some details remain unclear, we would be glad to clarify them.
>
> We hope that your positive opinion of the content of the paper with the improvement in clarity of presentation will merit an acceptance.

---

### Official Review · AnonReviewer2 · 2017-12-03
**Interesting direction but unconvincing experiments and uncompelling applications**

**Rating:** 4
**Confidence:** 3

**Review:**

This paper adds an interesting twist on top of recent unpaired image translation work. A domain-level translation function is jointly optimized with an instance-level matching objective. This yields the ability to extract corresponding image pairs out of two unpaired datasets, and also to potentially refine unpaired translation by subsequently training a paired translation function on the discovered matches. I think this is a promising direction, but the current paper has unconvincing results, and it’s not clear if the method is really solving an important problem yet.

My main criticism is with the experiments and results. The experiments focus almost entirely on the setting where there actually exist exact matches between the two image sets. Even the partial matching experiments in Section 4.1.2 only quantify performance on the images that have exact matches. This is a major limitation since the compelling use cases of the method are in scenarios where we do not have exact matches. It feels rather contrived to focus so much on the datasets with exact matches since, 1) these datasets actually come as paired data and, in actual practice, supervised translation can be run directly, 2) it’s hard to imagine datasets that have exact but unknown matches (I welcome the authors to put forward some such scenarios), 3) when exact matches exist, simpler methods may be sufficient, such as matching edges. There is no comparison to any such simple baselines.

I think finding analogies that are not exact matches is much more compelling. Quantifying performance in this case may be hard, and the current paper only offers a few qualitative results. I’d like to see far more results, and some attempt at a metric. One option would be to run user studies where humans judge the quality of the matches. The results shown in Figure 2 don’t convince me, not just because they are qualitative and few, but also because I’m not sure I even agree that the proposed method is producing better results: for example, the DiscoGAN results have some artifacts but capture the texture better in row 3.

I was also not convinced by the supervised second step in Section 4.3. Given that the first step achieves 97% alignment accuracy, it’s no surprised that running an off-the-shelf supervised method on top of this will match the performance of running on 100% correct data. In other words, this section does not really add much new information beyond what we could already infer given that the first stage alignment was so successful.

What I think would be really interesting is if the method can improve performance on datasets that actually do not have ground truth exact matches. For example, the shoes and handbags dataset or even better, domain adaptation datasets like sim to real.

I’d like to see more discussion of why the second stage supervised problem is beneficial. Would it not be sufficient to iterate alpha and T iterations enough times until alpha is one-hot and T is simply training against a supervised objective (Equation 7)?

Minor comments:
1. In the intro, it would be useful to have a clear definition of “analogy” for the present context.
2. Page 2: a link should be provided for the Putin example, as it is not actually in Zhu et al. 2017.
3. Page 3: “Weakly Supervised Mapping” — I wouldn’t call this weakly supervised. Rather, I’d say it’s just another constraint / prior, similar to cycle-consistency, which was referred to under the “Unsupervised” section.
4. Page 4 and throughout: It’s hard to follow which variables are being optimized over when. For example, in Eqn. 7, it would be clearer to write out the min over optimization variables.
5. Page 6: The Maps dataset was introduced in Isola et al. 2017, not Zhu et al. 2017.
6. Page 7: The following sentence is confusing and should be clarified: “This shows that the distribution matching is able to map source images that are semantically similar in the target domain.”
7. Page 7: “This shows that a good initialization is important for this task.” — Isn’t this more than initialization? Rather, removing the distributional and cycle constraints changes the overall objective being optimized.
8. In Figure 2, are the outputs the matched training images, or are they outputs of the translation function?
9. Throughout the paper, some citations are missing enclosing parentheses.

---

> ### Author Response · Authors · 2017-12-13
> **Response to the rest of the comments**
>
> We thank the reviewer for the extensive style and reference comments. They have been fixed in the revised version:
> 1. A definition of “analogy” for the present context added to intro.
> 2. Putin example removed for need of space.
> 3. “Weakly Supervised Mapping” previous work section removed and references merged for need of space.
> 4. Optimization variables have been explicitly added to equations.
> 5. Maps dataset citation was changed to Isola et al. 2017
> 6. Removed confusing comment: “This shows that the distribution matching is able to map source images that are semantically similar in the target domain.”
> 7. “This shows that a good initialization is important for this task.”: one way of looking at it, is that the exemplar loss optimizes the matching problem that we care about but is a hard optimization task. The two other losses are auxiliary losses that help optimization converge. Clarification added in text.
> 8. The results shown for inexact matching are as follows: For alpha iterations and ANGAN we show the matches recovered by our methods, The DiscoGAN results are the outputs of the translation function.
> 9. Parentheses added to all citations.
>
> We hope that this has convinced the reviewer of the importance of this work and are keen to answer any further questions.

---

> ### Author Response · Authors · 2017-12-13
> **Response to the motivation and experimental comments**
>
> Thank you for the detailed and constructive review. It highlighted motivation and experimental protocols that were further clarified in the revised version.
>
> This paper is focused on exact analogy identification. A core question in the reviews was the motivation for the scenario of exact matching, and we were challenged by the reviewer to find real world applications for it.
>
> We believe that finding exact matches is an important problem and occurs in multiple real-world problems. Exact or near-exact matching occurs in:
> * 3D point cloud matching.
> * Matching between different cameras panning the same scene in different trajectories (hard if they are in different modalities such as RGB and IR).
> * Matching between the audio samples of two speakers uttering the same set of sentences.
> * Two repeats of the same scripted activity (recipe, physics experiment, theatrical show)
> * Two descriptions of the same news event in different styles (at the sentence level or at the story level).
> * Matching parallel dictionary definitions and visual collections.
> * Learning to play one racket sport after knowing to play another, building on the existing set of acquired movements and skills.
>
> In all these cases, there are exact or near exact analogies that could play a major rule in forming unsupervised links between the domains.
>
> We note that on a technical level, most numerical benchmarks in cross domain translation are already built using exact matches, and many of the unsupervised techniques could be already employing this information, even if implicitly. We show that our method is more effective at it than other methods.
>
> On a more theoretical level, cognitive theories of analogy-based reasoning mostly discuss exact analogies from memory (see, e.g., G. Fauconnier, and M. Turner, “The way we think”, 2002 ). For example, a new situation is dealt with by retrieving and adopting a motor action that was performed before. Here, the chances of finding such analogies are high since the source domain is heavily populated due to life experiences.
>
> Regarding experiments. We believe that in some cases the requests are conflicting: we cannot provide numerical results in places for which there are no analogies and no metrics for success. We provide a large body of experiments for exact matches and show that our method far surpasses everything else. We have compared with multiple baselines covering all the reasonable successful approaches for matching between domains.
>
> The experiments regarding cases without exact matches are, admittedly, less extensive, added for completeness, and not the focus of this paper.
>
> The reviewer wondered if matching will likely work better with simpler methods. Our baselines test precisely this possibility and show that the simpler methods do not perform well. Specifically edge-based matches are well covered by the more general VGG feature baseline (which uses also low level maps - not just fc7). AN-GAN has easily outperformed this method. If it is possible to hand-craft a successful method for each task individually, these hand-crafted features are unlikely to generalize as well as the multi-scale VGG features or AN-GAN.
>
> We put further clarification in the paper for the motivation for the second “supervised” step. In unsupervised semantic matching, larger neural architecture have been theoretically and practically shown to be less successful (due to overfitting and finding it less easy to recover the correct transformation). The distribution matching loss function (e.g. CycleGAN) is adversarial and is therefore less stable and might not optimize the quantity we care about (e.g. L1/L2 loss). Once the datasets are aligned and analogies are identified, however, the cross domain translation becomes a standard supervised deep learning problem where large architectures do well and standard loss functions can be used. This is the reason for the two steps. It might be possible to include the increase in architecture into the alpha-iterations but it’s non-trivial and we didn’t find it necessary.

---

> > ### Comment · AnonReviewer2 · 2018-01-13
> > **Response to rebuttal**
> >
> > Thank you for your detailed reply.  I still think the paper could be much improved with more extensive experiments and better applications. However, I agree that the problem setting is interesting and novel, the method is compelling, and the experiments provide sufficient evidence that the method actually works. Therefore I would not mind seeing this paper accepted into ICLR, and, upon reflection, I think this paper is hovering around the acceptance threshold.
> >
> > I really like the real-world examples listed! I would be excited to see the proposed method applied to some of these problems. I think that would greatly improve the paper. (Although, I would still argue that several of the listed examples are cases where the data would naturally come in a paired format, and direct supervision could be applied.)
> >
> > It's a good point that previous unsupervised, cross domain GANs were also evaluated on contrived datasets with exact matches available at training time. However, I'd argue that these papers were convincing mainly because of extensive qualitative results on datasets without exact matches. Those qualitative results were enough to demonstrate that unpaired translation is possible. The current paper aims to go further, and show that the proposed method does _better_ at unpaired translation than previous methods. Making a comparison like this is harder than simply showing that the method can work at all, and I think it calls for quantitative metrics on real unpaired problems (like the examples listed in the rebuttal).
> >
> > There are a number of quantitative ways to evaluate performance on datasets without exact matches. First, user studies could be run on Mechanical Turk. Second, unconditional metrics could be evaluated, such as Inception score or moment matching (do the statistics of the output distribution match the statistics of the target domain?).
> >
> > However, I actually think it is fine to evaluate on ground truth matches as long as the training data is less contrived. For example, I would find it compelling if the system were tested on 3D point cloud matching, even if the training data contains exact matches, as long as there is no trivial way of finding these matches.

---

> > > ### Author Response · Authors · 2018-01-15
> > > **Additional experiment requested by Reviewer**
> > >
> > > We are deeply thankful to AnonReviewer2 for holding an open discussion and for acknowledging the significance of the proposed problem setting, the work’s novelty, and the quality of the experiments.
> > > We are also happy that AnonReviewer2 found the list of possible applications, provided in reply to the challenge posted in the review, to be exciting. We therefore gladly accept the new challenge that was set, to demonstrate the success of our method on one of the proposed applications in the list.
> > > Since the reviewer explicitly requested 3D point cloud matching, we have evaluated our method on this task. It should be noted that our method was never tested before in low-D settings, so this experiment is of particular interest.
> > > Specifically, we ran the experiment using the Bunny benchmark, exactly as is shown in “Discriminative optimization: theory and applications to point cloud registration”, CVPR’17 available as an extended version at https://arxiv.org/pdf/1707.04318.pdf, Sec. 6.2.3 . In this benchmark, the object is rotated by a random degree, and we tested the success rate of our model in achieving alignment for various ranges of rotation angles.
> > > For both CycleGAN and our method, the following architecture was used. D is a fully connected network with 2 hidden layers, each of 2048 hidden units, followed by BatchNorm and with Leaky ReLU activations. The mapping function is a linear affine matrix of size 3 * 3 with a bias term. Since in this problem, the transformation is restricted to be a rotation matrix, in both methods we added a loss term that encourages orthonormality of the weights of the mapper.  Namely, ||WW^T-I||, where W are the weights of our mapping function.
> > > The table below depicts the success rate for the two methods, for each rotation angle bin, where success is defined in this benchmark as achieving an RMSE alignment accuracy of 0.05.
> > > Rotation angle | CycleGAN |   Ours
> > > ============================
> > > 0-30                    0.12000     1.00000
> > > 30-60                  0.12500     1.00000
> > > 60-90                  0.11538     0.88462
> > > 90-120                0.07895     0.78947
> > > 120-150              0.05882     0.64706
> > > 150-180              0.10000     0.76667
> > >
> > > Comparing to the results reported in Fig. 3 of  https://arxiv.org/pdf/1707.04318.pdf, middle column, our results seem to significantly outperform the methods presented there at large angles. Therefore, the proposed method outperforms all baselines and, once again, proves to be effective as well as broadly applicable.
> > > P.S. It seems that the comment we posted above, which was titled “A real-world application of our method in cell biology” (https://openreview.net/forum?id=BkN_r2lR-&noteId=rJ6aA85QG), went unnoticed. In a way, it already addressed the new challenge by presenting quantitative results on a real-world dataset for which there are no underlying ground truth matches.

---

> > > > ### Comment · AnonReviewer2 · 2018-01-15
> > > > **New experiments are compelling, recommend accept**
> > > >
> > > > I thank the authors for thoroughly responding to my concerns. The 3D alignment experiment looks great, and indeed I did miss the comment about the cell bio experiment. That experiment is also very compelling.
> > > >
> > > > I think with these two experiments added to the revision, along with all the other improvements, the paper is now much stronger and should be accepted!

---

### Author Response · Authors · 2018-01-03
**A real-world application of our method in cell biology**

Two reviewers were concerned that the problem of unsupervised simultaneous cross-domain alignment and mapping, while well suited to the existing ML benchmarks, may not have real-world applications. In our rebuttal, we responded to the challenge posed by AnonReviewer2 to present examples of applications with many important use cases.

In order to further demonstrate that the task has general scientific significance, we present results obtained using our method in the domain of single cell expression analysis. This field has emerged recently, due to new technologies that enable the measurement of gene expression at the level of individual cells. This capability already led to the discovery of quite a few previously unknown cell types and holds the potential to revolutionize cell biology. However, there are many computational challenges since the data is given as sets of unordered measurements. Here, we show how to use our method to map between gene expression of cell samples from two individuals and find interpersonal matching cells.

From the data of [1], we took the expressions of blood cells (PMBC) extracted for donors A and B (available online at https://support.10xgenomics.com/single-cell-gene-expression/datasets; we used the matrices of what is called “filtered results”). These expressions are sparse matrices, denoting 3k and 7k cells in the two samples and expressions of around 32k genes.  We randomly subsampled the 7k cells from donor B to 3k and reduced the dimensions of each sample from 32k to 100 via PCA. Then, we applied our method in order to align the expression of the two donors (find a transformation) and match between the cell samples in each. Needless to say, there is no supervision in the form of matching between the cells of the two donors and the order of the samples is arbitrary. However, we can expect such matches to exist.

We compare three methods:
The mean distance between a sample in set A and a sample in set B (identity transformation).
The mean distance after applying a CycleGAN to compute the transformation from A to B (CG for CycleGAN).
The mean distance after applying our complete method.

The mean distance with the identity mapping is 3.09, CG obtains 2.67, and our method 1.18. The histograms of the distances are shown in the anonymous url:
https://imgur.com/xP3MVmq

We see a great potential in further applying our method in biology with applications ranging from interspecies biological network alignment [2] to drug discovery [3], i.e. aligning expression signatures of molecules to that of diseases.

[1] Zheng et al, “Massively parallel digital transcriptional profiling of single cells”. Nature Communications, 2017.

[2] Singh, Rohit, Jinbo Xu, and Bonnie Berger. "Global alignment of multiple protein interaction networks with application to functional orthology detection." Proceedings of the National Academy of Sciences 105.35 (2008): 12763-12768.

[3] Gottlieb, et al. "PREDICT: a method for inferring novel drug indications with application to personalized medicine." Molecular systems biology 7.1 (2011): 496.

---

### Decision · Program_Chairs · 2018-01-29
**ICLR 2018 Conference Acceptance Decision**

**Decision:**

Accept (Poster)

**Comment:**

This paper builds on top of Cycle GAN ideas where the main idea is to jointly optimize the domain-level translation function with an instance-level matching objective. Initially the paper received two negative reviews (4,5) and a positive (7). After the rebuttal and several back and forth between the first reviewer and the authors, the reviewer was finally swayed by the new experiments. While not officially changing the score, the reviewer recommended acceptance. The AC agrees that the paper is interesting and of value to the ICLR audience.